METHODS

# TastepepAI: An artificial intelligence platform for taste peptide *de novo* design

**Jianda Yue**[1,2,3☯], **Tingting Li**[1,2,3☯], **Jian Ouyang**[4,5], **Jiawei Xu**[1,2,3], **Hua Tan**[1,2,3], **Zihui Chen**[1,2,3], **Changsheng Han**[1,2,3], **Huanyu Li**[1,2,3], **Songping Liang**[1,2,3], **Zhonghua Liu**[1,2,3]*, **Zhonghua Liu**[1,2,3,4,5]*, **Ying Wang** [1,2,3]*

**1** The National and Local Joint Engineering Laboratory of Animal Peptide Drug Development, College of Life Sciences, Hunan Normal University, Changsha, Hunan, China, **2** Peptide and small molecule drug R&D plateform, Furong Laboratory, Hunan Normal University, Changsha, Hunan, China, **3** Institute of Interdisciplinary Studies, Hunan Normal University, Changsha, Hunan, China, **4** Key Laboratory of Tea Science of Ministry of Education, Hunan Agricultural University, Changsha, China, **5** National Research Center of Engineering Technology for Utilization of Functional Ingredients from Botanicals, Hunan Agricultural University, Changsha, China

☯ The authors give equal contributions
* liuzh@hunnu.edu.cn (ZL); zhonghua-liu-ms@hunau.edu.cn (ZL); wangyin@hunnu.edu.cn (YW)

## Abstract

Taste peptides have emerged as promising natural flavoring agents attributed to their unique organoleptic properties, high safety profile, and potential health benefits. However, the *de novo* identification of taste peptides derived from animal, plant, or microbial sources remains a time-consuming and resource-intensive process, significantly impeding their widespread application in the food industry. In this work, we present TastePepAI, a comprehensive artificial intelligence framework for customized taste peptide design and safety assessment. As the key element of this framework, a loss-supervised adaptive variational autoencoder (LA-VAE) is implemented to efficiently optimize the latent representation of sequences during training and facilitate the generation of target peptides with desired taste profiles. Notably, our model incorporates a novel taste-avoidance mechanism, allowing for selective flavor exclusion. Subsequently, our in-house developed toxicity prediction algorithm (SpepToxPred) is integrated in the framework to undergo rigorous safety evaluation of generated peptides. Using this integrated platform, we successfully identified 73 peptides exhibiting sweet, salty, and umami, significantly expanding the current repertoire of taste peptides. This work demonstrates the potential of TastePepAI in accelerating taste peptide discovery for food applications and provides a versatile framework adaptable to broader peptide engineering challenges.

**Data availability statement:** Data availability All taste peptides investigated in this study have been deposited in our established taste peptide database, TastePepMap (http://www.wang-subgroup.com/TastePepMap.html), which is freely accessible to the research community. Code availability The web servers TastePepMap, TastePepAI, and SpepToxPred developed in this study are freely accessible at http://www.wang-subgroup.com/TastePepMap.html, http://www.wang-subgroup.com/TastePepAI.html , and http://www.wang-subgroup.com/TasToxPred/TasToxPred.html, respectively. The source codes are made available to the community of researchers and developers at https://github.com/leleshidawang/TastepepAI .

**Funding:** This work was supported in part by the National Natural Science Foundation of China (Grants No. 32171271 to S.L., 32271329 to Z.L.(刘中华), and 22473041 to Y.W.), the Natural Science Foundation of Hunan Province (Grant No. 2024JJ2042 to Y.W) and Scientific research project of Education Department of Hunan Province (Key Project, Grant No. 23A0084 to Y.W.). The funders had no role in study design, data collection and analysis, decision to publish, or preparation of the manuscript.

**Competing interests:** The authors have declared that no competing interests exist.

## Author summary

Taste peptides have established themselves as attractive natural flavor enhancers, thanks to their distinct sensory attributes, strong safety record, and possible health advantages. TastePepAI, the first artificial intelligence platform for designing taste peptides with desired flavor profiles, was developed in this work. Traditional methods for identifying taste peptides are time-consuming and costly, with their applications in the food industry limited. Two key innovations are featured in our integrated computational framework: LA-VAE, which is used for generating peptide sequences with target taste properties while suppressing unwanted characteristics, and SpepToxPred for safety assessment—with its accuracy being 12% higher than that of existing toxicity prediction models. Using this platform, 73 novel multifunctional taste peptides exhibiting sweet, salty, and umami properties were successfully designed and validated. Electronic tongue analysis confirmed their expected taste characteristics, while safety assays demonstrated excellent biocompatibility. To promote open science, we established the TastePepMap database and TastePepAI design platform. This work demonstrates AI's potential in functional peptide design and provides crucial methodological foundations for developing next-generation peptide-based taste modulators, offering new opportunities for creating healthier and more sustainable food ingredients.

## 1. Introduction

Taste perception fundamentally influences food selection and consumption behavior [1,2]. Taste peptides, emerging as natural taste-modulating compounds, have attracted considerable attention [3,4]. These bioactive peptides, comprising 2–20 amino acid residues [5,6], can trigger multiple taste perceptions including sweet, umami, and salty tastes, without the drawbacks of conventional flavoring agents [5,7]. They offer distinct advantages: easily metabolized and absorbed due to their natural amino acid composition [8,9], simultaneous multiple taste modalities reducing the need for various flavorings [10,11], and additional health-promoting functions such as antioxidization [12,13] and anti-inflammatory [14,15] properties, unlike traditional flavoring agents (such as sodium chloride, sucrose, and monosodium glutamate) that may cause health issues [16–19]. This combination of sensory enhancement and health benefits presents a promising solution to the palatability-health paradox in modern food industry [20,21].

Advances in taste peptide research have demonstrated significant applications. In salt reduction, specific peptides maintain sensory qualities while reducing NaCl content in meat products [22,23]. A decapeptide (0.4 g/L) from fermented tofu enhances 50 mM NaCl perception to 63 mM equivalent [24]. And a peptide from Ruditapes philippinarum hydrolysate elevates 3 g/L NaCl perception to 5 g/L equivalent [25,26]. For sugar reduction, sweet peptides like aspartame [27] and neotame [28] are widely

used, with aspartame showing additional anti-inflammatory benefits [29,30]. Sweet peptides from mulberry seed protein demonstrate six-fold higher sweetness than 0.1 g/mL sucrose [31]. Umami peptides from various food sources enhance flavor while reducing salt and monosodium glutamate usage through synergistic interactions [11,23,32].

However, traditional identification methods for taste peptides face substantial challenges [33,34]. The conventional workflow of preprocessing, extraction, purification, synthesis, and sensory evaluation remains time-consuming and resource-intensive, yielding limited peptide samples [35,36]. Furthermore, biological sample complexity and experimental variations may result in limitations in taste peptide applications, including toxicity risks [37,38], stability issues [39], or inadequate taste properties [40], significantly increasing development costs and complexity.

To address these experimental challenges, various computational and artificial intelligence-based approaches for taste peptide identification have emerged. Most reported methods focus on predicting single taste properties, such as BERT-4Bitter [41] and iBitter-SCM [42] for bitter peptide identification, while iUmami-SCM [43], Umami-MRNN [44, iUmami-DRLF [45], and Umami-gcForest [46] are dedicated to umami peptide prediction. Additionally, a few models like Umami_YYDS [35] have been reported to perform binary classification for umami and bitter tastes. However, current methodologies remain largely rudimentary for several reasons. First, existing taste peptide prediction models have limited complexity, and their capability to identify at most two tastes falls short of practical requirements [35,41–46], as peptides may exhibit multiple taste modalities (sour, sweet, bitter, salty, umami). Second, data processing presents significant limitations. Many current prediction studies construct datasets that position bitter taste as the antithesis of umami, creating a binary classification framework [35,46]. However, many peptides, such as GEG [47,48], EGF [47] and KGDEESLA [47,49], are known to possess both umami and bitter characteristics. This suggests that different tastes typically coexist within a same peptide rather than separate in a binary state. Furthermore, relying solely on prediction models may offer less assistance in accelerating potential taste peptide discovery than anticipated, as they merely classify existing sequences as 'positive' or 'negative' without the capability to generate novel sequences.

In this work, we present TastePepAI, the first integrated artificial intelligence (AI) platform for *de novo* design and evaluation of taste peptides (Fig 1). At its core, the platform features the loss-supervised adaptive variational autoencoder (LA-VAE) that achieves precise modeling of high-dimensional sequence spaces through dynamic optimization of reconstruction loss ($Loss_{rec}$) and Kullback-Leibler (KL) divergence ($Loss_{KL}$) during the encoding-decoding process. Notably, this platform introduces a taste-avoidance strategy enabling the model to generate desired taste properties while suppressing unwanted taste characteristics. To ensure the safety of generated sequences, we implemented SpepToxPred, an evaluation model based on sequence-toxicity relationships. Using this AI computational framework, 73 peptides with sweet, salty, and umami tastes have been successfully discovered and then experimentally evaluated, exceeding the total number of previously reported peptides with these three taste characteristics and significantly expanding the known sequence spaces of taste peptides. Experimental validation also confirmed that these peptides showed no significant toxicity toward mammalian red blood cells and non-cancerous cells. All sequence data have been made publicly available in our taste peptide database, TastePepMap. Given its performance in both computational and experimental evaluations, TastePepAI represents a significant advancement in the AI-driven design and development of taste peptides. Fig 1 shows the Taste-PepAI platform for taste peptide design.

## 2. Results

### 2.1. Comprehensive analysis reveals the sequence characteristics and complex taste properties of taste peptides

The taste peptides demonstrate a clear predominance of short sequences, with 88.54% of the collected peptides not exceeding 10 amino acids in length, while sequences of 15 amino acids or longer constitute less than 3% (Fig 2A). Regarding taste type distribution, the curated dataset exhibits significant imbalance: umami (575) and bitter (541) peptides dominate, while sour (201), sweet (162), and salty (141) peptides are relatively underrepresented (Fig 2B). This

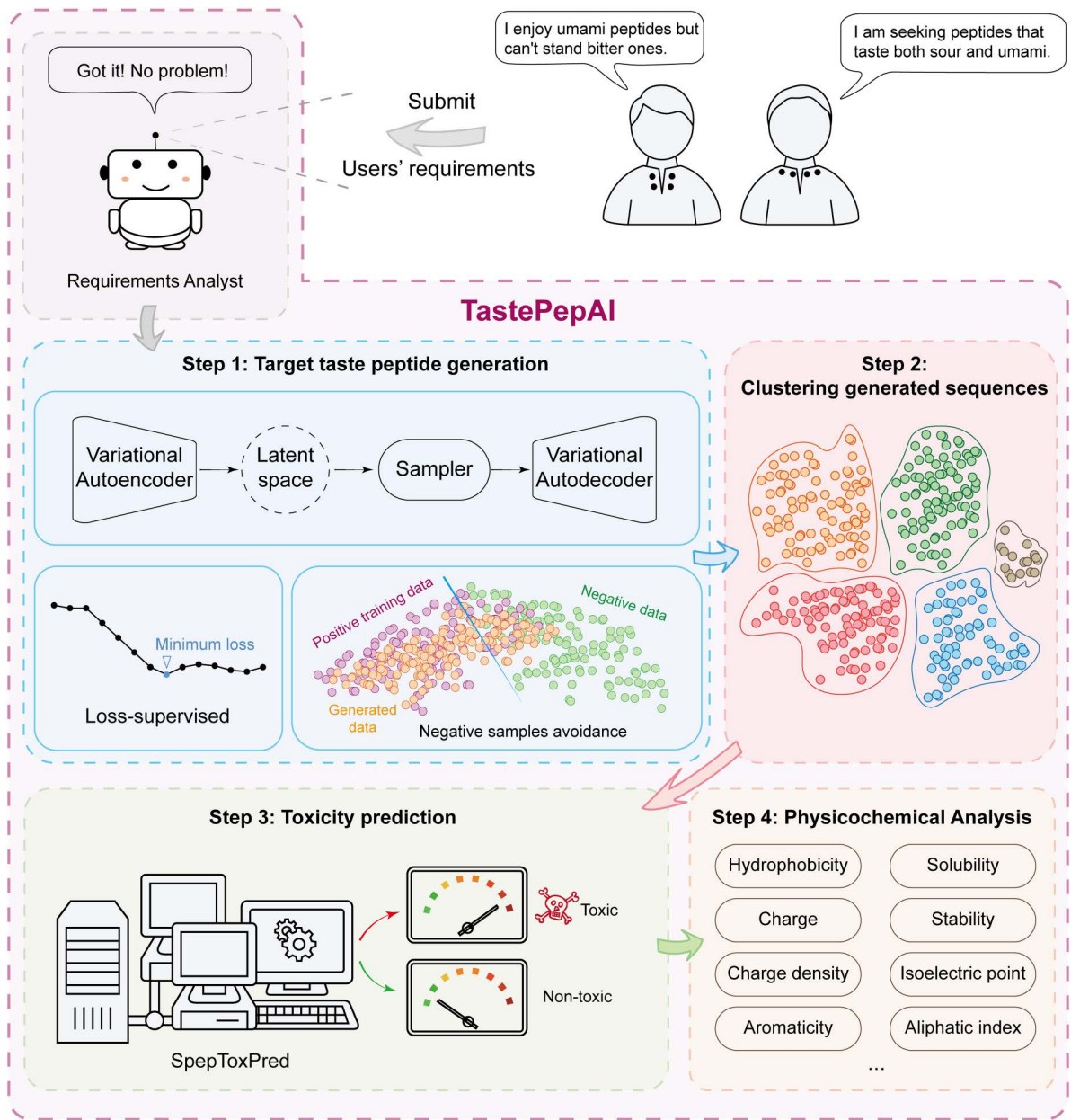

**Fig 1. Overview of the TastePepAI platform for taste peptide design.** TastePepAI is a fully automated integrated computational platform. The platform firstly analyzes users' requirements for specific taste characteristics, followed by four main steps: (1) Target taste peptide generation (light blue panel): utilizing LA-VAE to generate sequences with desired taste properties while suppressing unwanted taste. (2) Clustering analysis of generated sequences to select representative peptide sequences (light red panel). (3) Toxicity prediction using SpepToxPred (yellow-green panel). (4) Comprehensive physicochemical analysis of candidate peptides, including properties such as hydrophobicity, solubility, charge, stability, charge density, isoelectric point, aromaticity, and aliphatic index (light yellow panel).

distribution pattern likely stems from two key factors: the application potential of umami peptides in food flavor enhancer development has driven related research [20,34,50], while the preferential generation mechanism of bitter peptides during protein proteolysis leads to their prevalence in fermented products [51,52]. Amino acid composition analysis (Fig 2C)

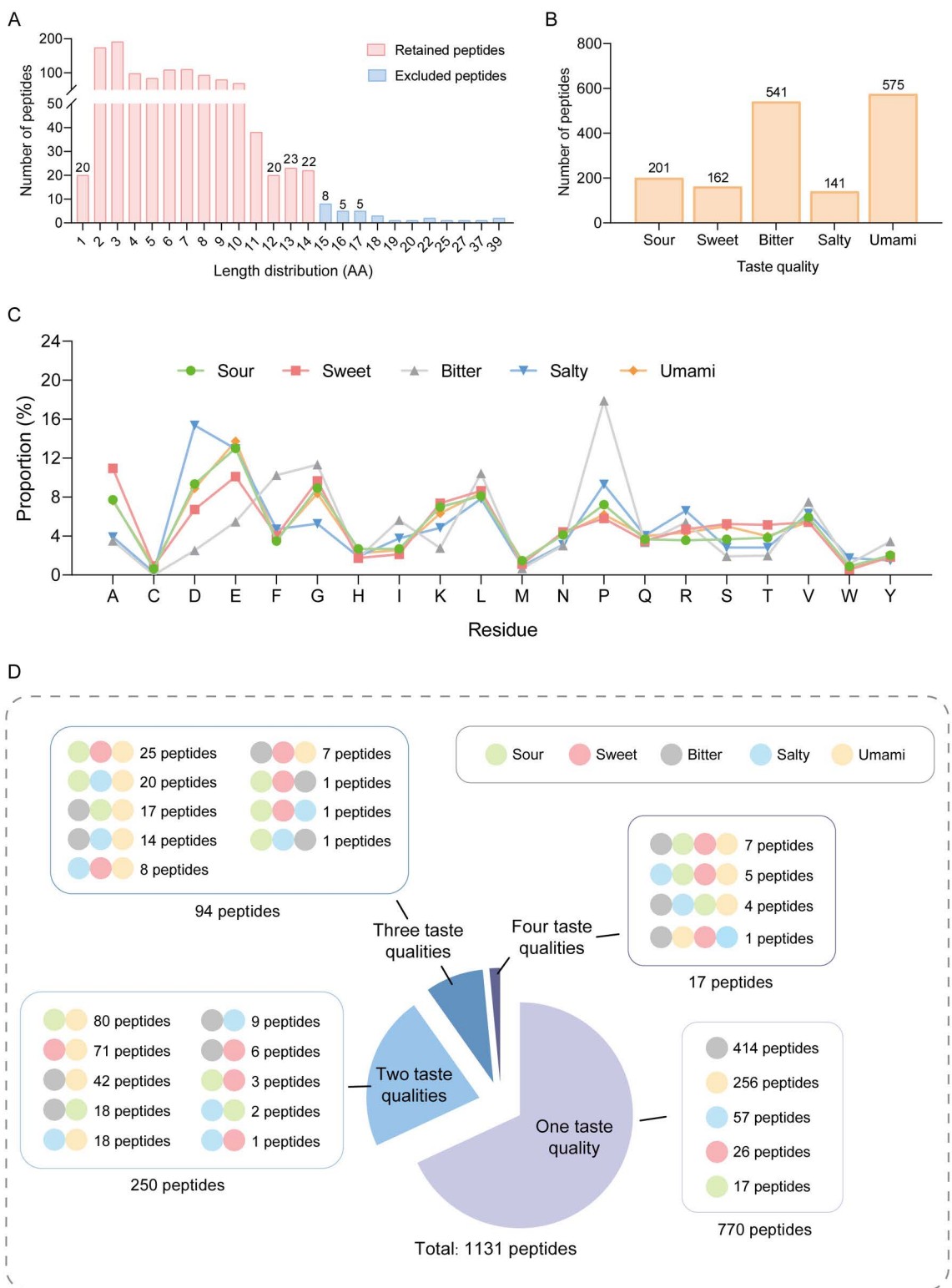

**Fig 2. Sequence characteristics and taste property distribution of the curated taste peptide dataset. (A)** Length distribution of taste peptides. **(B)** Distribution of peptides across five basic taste categories: sour (201), sweet (162), bitter (541), salty (141), and umami (575) peptides. **(C)** Amino acid composition analysis across different taste categories. **(D)** Non-redundant classification analysis of 1131 taste peptides revealing the distribution of

single and multiple taste properties. Colored circles represent different tastes (sour: light green, sweet: light red, bitter: light gray, salty: light blue, umami: light yellow), with multiple circles indicating peptides possessing multiple taste properties.

reveals diverse residue distribution patterns among different taste peptides, with sour and umami peptides showing high compositional similarity, while salty and bitter peptides are distinguished by high frequencies of aspartic acid (15.36%) and proline (17.90%), respectively.

Non-redundant classification analysis (Fig 2D) further unveils the complexity of taste peptides, showing that among 1131 peptides, over 30% possess multiple taste characteristics beyond single-taste peptides (770). Dual-taste peptides (250) exhibit rich combination patterns, including sour-umami (80), sweet-umami (71), and bitter-umami (42); triple-taste peptides (94) form more complex combinations such as sweet-sour-umami (25) and sour-salty-umami (20); some peptides even possess four taste properties (17). Additionally, sequence alignment results (Fig A-A and A-B in S1 Text) indicate high heterogeneity within the same taste category, encompassing both highly similar peptide clusters and groups with substantial sequence variations. Moreover, peptides of different taste categories show highly overlapping distribution patterns in similarity space without clear boundaries. This complex sequence-function relationship not only reflects the sequence diversity of taste peptides but also suggests the flexibility and complexity of taste recognition mechanisms.

## 2.2. LA-VAE: a loss-supervised adaptive variational autoencoder with contrastive learning for controlled taste peptide generation

To address the intricate sequence-function relationships of taste peptides, we developed LA-VAE as the core algorithm of TastePepAI. LA-VAE introduces an innovative dynamic loss supervision mechanism that enables precise control over the model training process (Fig 3A). This mechanism partitions the training process into two complementary optimization phases: an initial exploration phase (Phase I) that continuously tracks and records the global optimal loss, and a convergence optimization phase (Phase II) that captures optimal latent space representations through strict dual constraints (simultaneous decrease in both $Loss_{rec}$ and $Loss_{KL}$). This loss-aware adaptive optimization framework not only provides an effective model state capture mechanism but also establishes a dynamically balanced quality control system for sequence generation. Notably, when the convergence phase fails to achieve expected performance improvements, LA-VAE automatically triggers an elastic extension mechanism that continues to explore optimal solutions through configurable extension cycles, thereby ensuring the reliability of generated sequences. This multi-phase collaborative optimization strategy significantly enhances both model convergence efficiency and generation quality.

To meet the demands for precise control of specific taste properties (e.g., bitter taste elimination [40,53]) in practical applications, we incorporated a contrastive learning-based taste avoidance mechanism into the LA-VAE architecture (Fig 3B). Our contrastive approach differs from traditional contrastive learning methods that use explicit contrastive loss functions. Instead, it implements an implicit contrastive mechanism through bilateral data partitioning and latent space distance evaluation. This mechanism allows users to explicitly specify desired (preferred) and avoided (aversive) taste properties, enabling bilateral partitioning of training data: sequences with target taste features constitute the positive set, while those with avoided features form the negative set. Through parallel training on these complementary datasets, LA-VAE establishes a structured contrastive representation framework in the latent space. During sequence generation, the model employs a k-nearest neighbor (k = 5) based bilateral distance evaluation strategy, computing the average Euclidean distances between each candidate sequence and both positive and negative sample sets in the latent space. Sequences that simultaneously satisfy both "positive sample affinity" and "negative sample repulsion" criteria are selected as final outputs. This precise screening mechanism, based on latent space topology, not only guarantees the target properties of generated sequences but also establishes an effective taste feature control system.

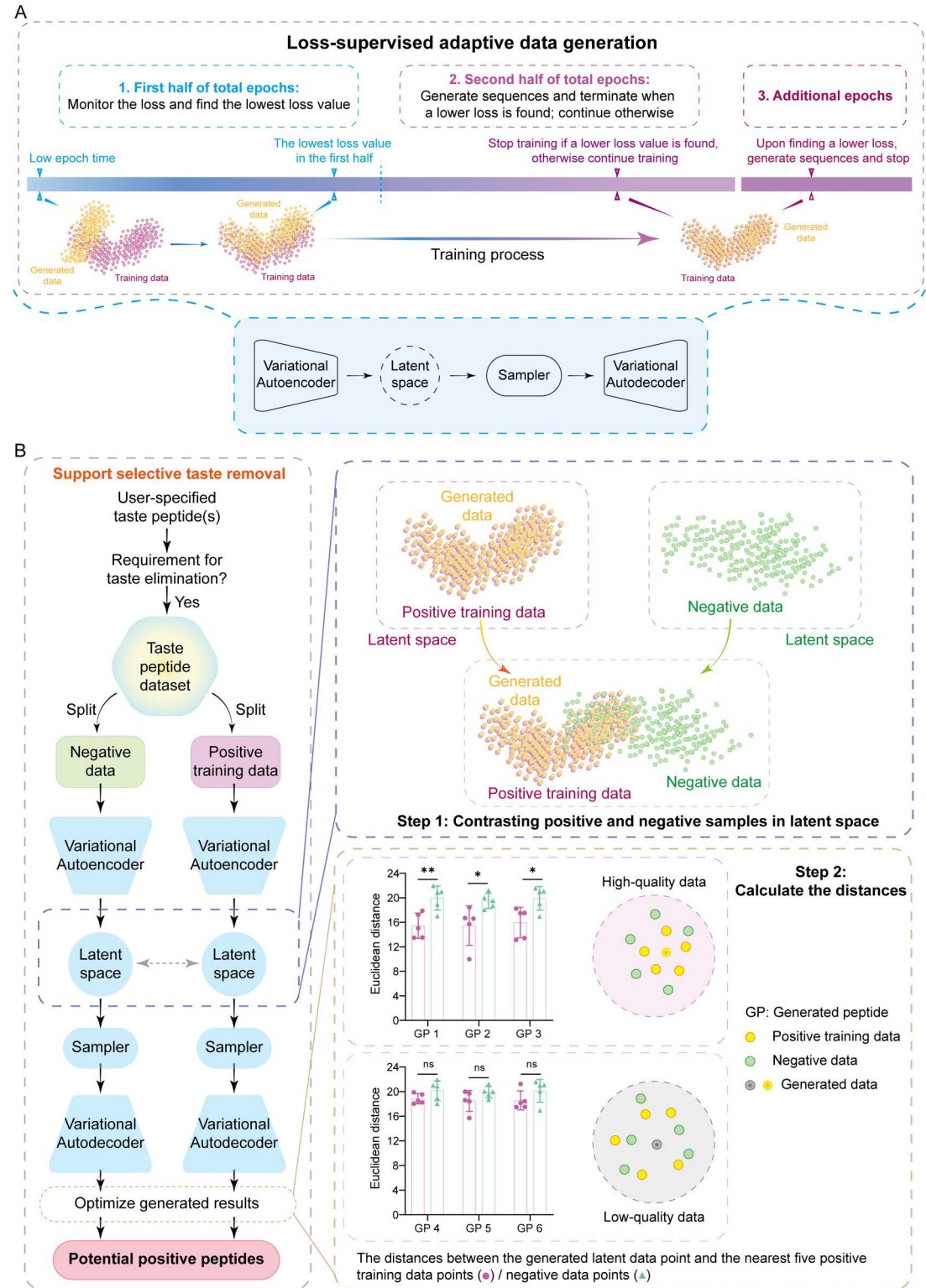

**Fig 3. Architecture and workflow of LA-VAE. (A)** Schematic illustration of the loss-supervised adaptive data generation framework. The training process is strategically divided into three phases: (1) Initial exploration phase (first half of total epochs, blue) monitors and records the global minimum loss while maintaining the model's generative capability; (2) Convergence optimization phase (second half of total epochs, purple) generates sequences

and terminates upon discovering a lower loss value, otherwise continues training; (3) Extension phase (additional epochs, dark purple) activates when a new optimal loss is not found during phase II, enabling further optimization. The lower panel shows the core components of the variational autoencoder architecture, including the encoder for latent space mapping, the latent space sampler, and the decoder for sequence reconstruction. Yellow and purple dots represent generated and training data points, respectively, illustrating the progressive refinement of the model's generative distribution. **(B)** Contrastive learning-based taste property control mechanism. Left panel: Workflow of selective taste removal, where user-specified taste peptides are split into positive training and negative sets, each processed through variational autoencoders to establish contrasting latent spaces. Middle panel (Step 1): Visualization of latent space distribution displaying positive training data (pink), negative data (green), and generated data points (orange). Right panel (Step 2): Quality assessment of generated peptides based on Euclidean distances to k-nearest neighbors (k = 5). Upper plots show high-quality generated peptides (GP 1-3) with significant distance differences between positive and negative samples (*p < 0.05, **p < 0.01), while lower plots demonstrate low-quality peptides (GP 4-6) with non-significant differences (ns). Scatter plots illustrate the spatial distribution of high-quality (upper) and low-quality (lower) generated peptides (gray) relative to positive training data (yellow) and negative data (green) in the latent space.

## 2.3. SpepToxPred: a specialized short peptide toxicity predictor

SpepToxPred, another core component of TastePepAI, was developed to evaluate the toxicity (including hemolytic activity [54], neurotoxicity [55], etc.) of sequences generated by LA-VAE. Initial analysis revealed that sequences under 50 amino acids constituted 99.39% of the total samples (Fig B-A in S1 Text), with cysteine (C) showing the highest frequency (>12%) in positive samples (Fig B-B in S1 Text). Given the primary application of TasToxPred in taste peptide toxicity prediction, we implemented a stringent length filtering strategy, retaining only sequences ≤25 amino acids for model development. This strategy was supported by four key findings: Firstly, after removing sequence redundancy, 2821 positive samples ≤ 25 amino acids remained (Fig B-C in S1 Text), hereafter referred to as shorter toxic peptides), providing sufficient statistical power for model training and validation. Secondly, the amino acid distribution patterns of shorter toxic/nontoxic peptides closely matched those of the complete dataset (Figs B-B and B-D in S1 Text), confirming the representative characteristics of the filtered dataset. Thirdly, length-specific residue frequency analysis of original positive sequences (Fig C in S1 Text) revealed more pronounced length-dependent fluctuations in key residues (e.g., C, K, L, R, W) among shorter toxic peptides, highlighting unique compositional patterns in short sequences. Finally, comparative analysis demonstrated significant differences in the occurrence frequencies of 18 amino acid residues between shorter and longer toxic peptides (Fig D in S1 Text), further supporting the necessity of length-specific modeling approaches.

To construct a high-precision toxicity prediction model, we designed a systematic feature engineering and model optimization framework (Fig 4A). This framework integrates 20 sequence encoding descriptors and 9 machine learning algorithms. The feature engineering phase established a multidimensional sequence feature space encompassing amino acid composition (AAC, DPC, etc.), physicochemical properties (CTDC, etc.), sequence encoding (CKSAAP, etc.), evolutionary information (BLOSUM62), and advanced features (DDE, Z-scale, etc.). We implemented a multi-stage feature selection strategy: first quantitatively evaluating single features and all possible dual-feature combinations, then employing iterative forward selection to construct more complex feature combinations. The algorithm assessed performance improvements upon adding each remaining feature to the current optimal feature set until no significant enhancement was observed.

In the algorithm optimization phase, comprehensive evaluation based on Matthews Correlation Coefficient (MCC) showed that Random Forest (RF) achieved optimal performance (MCC = 0.6231, accuracy = 0.8099, precision = 0.8483) with the BLOSUM62 + CTDD+DPC + AAC feature combination. We further developed a weighted voting-based ensemble learning framework, where the weight configuration (RF: 0.3, LGBM: 0.1, XGB: 0.2, KNN: 0.2, LR: 0.2) maximized prediction performance (MCC = 0.6540, accuracy = 0.8255, precision = 0.8629). The highest weight (0.3) of RF in the ensemble model indicated its superior capability in capturing toxicity-related sequence features (Fig 4A). Detailed weight calculation results (step size: 0.1) are available in S1 Data.

Systematic comparison with 13 existing peptide toxicity prediction tools on the independent test set (Fig 4B) demonstrated that the five optimal model configurations (SpepToxPred and Models 2–5) selected through 10-fold cross-validation exhibited consistently high performance. SpepToxPred achieved an MCC of 0.7019, representing a 12.79%

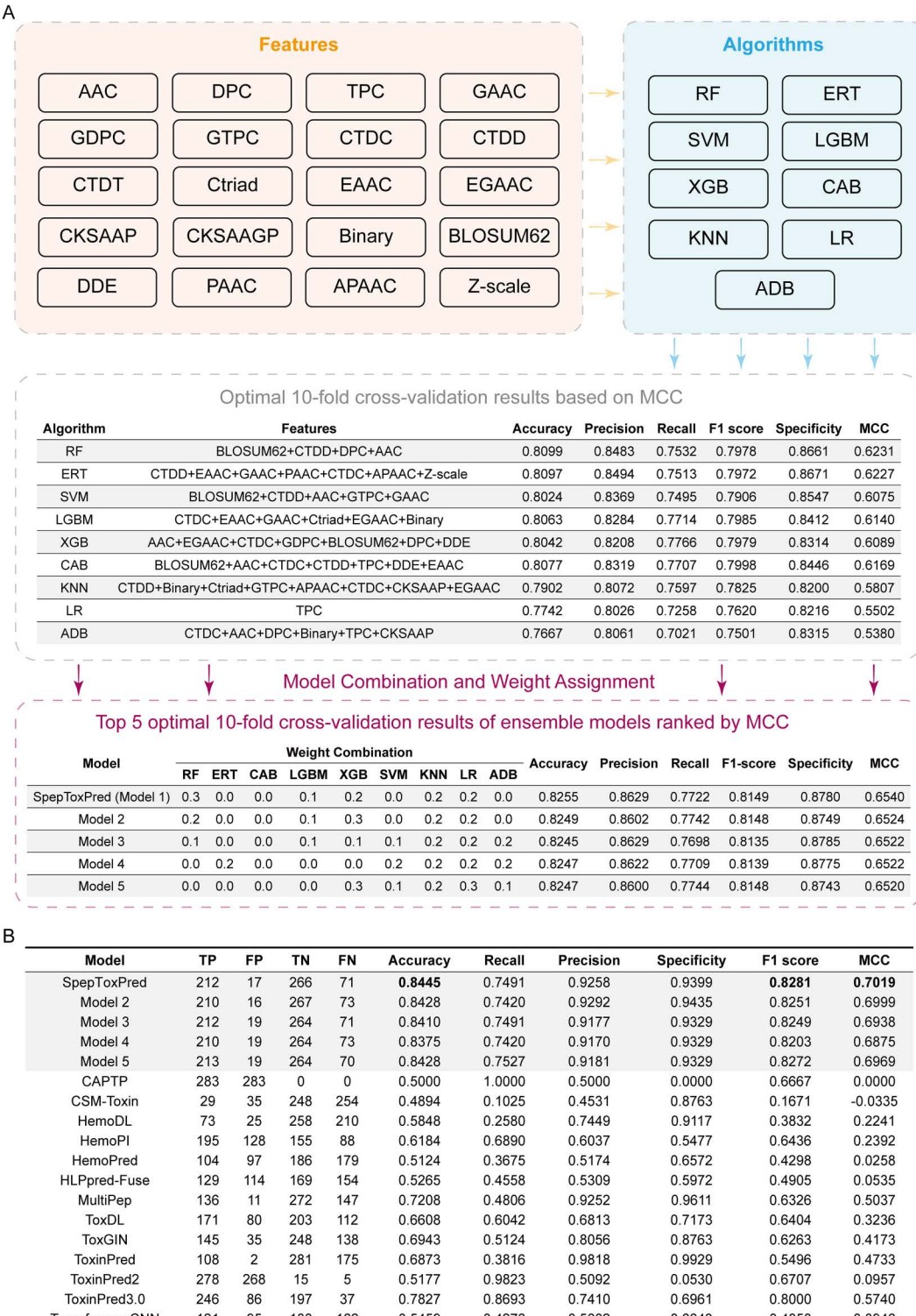

**Fig 4. Development and optimization of SpepToxPred. (A)** Systematic framework for feature engineering and model optimization. Upper panel: Integration of 20 sequence encoding descriptors (light yellow box) and 9 machine learning algorithms (light blue box). Middle panel: Performance evaluation of individual algorithms with their optimal feature combinations through 10-fold cross-validation, ranked by Matthews Correlation Coefficient (MCC). Lower panel: Weight optimization results for ensemble models, showing the top 5 configurations with different algorithm combinations. SpepToxPred

(Model 1) achieved optimal performance with weights distributed across RF (0.3), LGBM (0.1), XGB (0.2), KNN (0.2), and LR (0.2). Full spelling of the abbreviations of the features and algorithms are listed in Section 4.2.4. **(B)** Comprehensive performance comparison of SpepToxPred with 17 existing toxicity prediction tools on the independent test set. The evaluation metrics include true positives (TP), false positives (FP), true negatives (TN), false negatives (FN), accuracy, recall (sensitivity), precision, specificity, F1 score, and MCC. SpepToxPred and Models 2-5 represent the top five ensemble configurations from the optimization framework.

improvement over the best existing model ToxinPred 3.0 (MCC = 0.5740), with accuracy (0.8445) increased by 6.18%. Additionally, SpepToxPred demonstrated exceptional performance in precision and specificity (0.9258 and 0.9399, respectively). These superior performance metrics not only validate our feature engineering and model optimization strategies but also highlight the specialization and practical value of SpepToxPred in short peptide toxicity prediction.

### 2.4. Design and validation of safe taste peptides via TastePepAI

To demonstrate the practical utility of TastePepAI, we designed a challenging case study targeting the development of safe taste modulators. Given the health concerns associated with conventional taste enhancers, there is significant value in developing safe alternatives for sweet [31], salty [3], and umami [23,34] tastes. We therefore employed Taste-PepAI to generate novel sequences exhibiting these three basic tastes (individually or in combination) while maintaining safety and reducing bitterness. To achieve this, the training dataset was constructed by using peptides with target taste properties (sweet, salty, and umami, single or combined) without bitterness as positive samples, while bitter peptides served as negative samples for LA-VAE training. This multi-taste positive sampling strategy effectively captured the diversity and complexity of taste-sequence relationships, enabling comprehensive learning of sequence-taste association patterns.

To gain deeper insights into the training dynamics and optimization process of LA-VAE, we implemented a comprehensive training cycle of 500 epochs and monitored three critical time points (Figs E-A and E-B in S1 Text). During the early phase (Step 1, epoch 10), although the total loss remained relatively high ($Loss_{tol} = 7.5719$ for positive training and $Loss_{tol} = 0.7356$ for negative training), the model demonstrated rapid convergence. In the first-half optimization phase (Step 2), the model achieved initial optimal performance ($Loss_{tol} = 0.2088$ for positive training and $Loss_{tol} = 0.1720$ for negative training), followed by the global optimization phase (Step 3) where even lower loss values were attained ($Loss_{tol} = 0.1817$ for positive training and $Loss_{tol} = 0.1658$ for negative samples). Notably, both $Loss_{rec}$ and $Loss_{KL}$ exhibited synchronized reduction throughout the training process. The model's representational capacity showed progressive enhancement, with generated samples initially displaying distinct separation from positive samples in the latent space (Step 1, Fig E-C in S1 Text), followed by gradual convergence and substantial overlap with the positive sample distribution (Step 2, 3, Figs E-D and E-E in S1 Text). Intriguingly, while positive (purple) and negative (green) samples showed clear separation in the early training phase (Step 1), this distinct boundary gradually diminished as training progressed (Steps 2 and 3). This phenomenon further emphasizes the inherent overlapping characteristics and complexity of taste peptide sequences, highlighting the importance of our developed contrastive learning-based taste control mechanism that relies on Euclidean distance calculations.

Characteristic analysis of the generated sequences demonstrated that LA-VAE not only successfully captured key features of target sequences but also exhibited remarkable innovation. Specifically, the amino acid distribution of generated sequences maintained high similarity with positive samples (Fig F-A in S1 Text), reflecting the model's accurate learning of amino acid compositional patterns. Sequence similarity analysis revealed that the majority of generated peptides showed less than 50% sequence identity to positive samples (Fig F-B in S1 Text), strongly confirming the model's capability for deep feature extraction and novel sequence recombination rather than simple template copying or minor modifications. Furthermore, these novel sequences effectively preserved critical physicochemical properties of positive samples, including charge distribution, hydrophobicity, and hydrophobic ratio parameters (Figs F-C to F-L in S1 Text).

Through TastePepAI's automated screening pipeline, we manually selected 73 candidate peptides for experimental validation. These peptides were synthesized by Fmoc-solid peptide synthesis, and their purity exceeded 98%. All peptides exhibited excellent safety profiles at 100 μM, maintaining cell viability above 90% (Fig G-A in S1 Text) and hemolysis rates below 1.5% (Fig G-B in S1 Text). Electronic tongue analysis (Fig 5 and Tables A and B in S1 Text) revealed complex taste characteristics: at 0.1 mg/mL, all peptides demonstrated sweet and umami properties; at 1 mg/mL, salty characteristics emerged universally. Notably, certain peptides (e.g., TaPep8–11) exhibited relatively strong sweet and umami intensities at low concentrations while displaying different salty intensities at high concentrations, reflecting the complex dynamics of taste perception. Additionally, only few peptides (e.g., TaPep5, TaPep7, TaPep10) showed bitterness suppression at the high concentration (1 mg/mL). This limited bitterness suppression might be attributed to residual impurities from peptide synthesis (e.g., trifluoroacetic acid, sodium acetate) potentially interfering with sensory evaluation [10,56,57], coupled with possible taste evaluation blind spots in training data labeled as 'currently without bitterness' [58,59]. Consequently, accurate assessment of bitterness suppression requires further validation.

Nevertheless, TastePepAI successfully designed and validated 73 novel functional peptides with multiple target taste properties (sweet, salty, and umami), significantly expanding the existing taste peptide library. These results not only validate TastePepAI's technical advantages in direct taste peptide design but also provide crucial molecular foundations and methodological references for developing next-generation peptide-based taste modulators.

### 2.5. Online deployment of open-access tools for taste peptide research

To promote open sharing and practical applications in taste peptide research, we developed three interconnected yet functionally distinct platforms. First, we established TastePepMap (Fig 6A), a comprehensive taste peptide database currently hosting over 1200 sequences with professional curation mechanisms for regular updates through continuous monitoring of global taste peptide research advances. TastePepMap supports multidimensional queries, enabling users to flexibly retrieve detailed peptide information through sequence or taste characteristic searches (Fig 6B).

Second, addressing the demands from both academic and industrial sectors for taste peptide development, we launched the TastePepAI service platform (Fig 6C). This platform enables users to precisely define target taste characteristics and features to be avoided, offering two carefully designed training modes. 'Single Pattern Mode' focuses on precise training for specific taste combinations (e.g., for sour-sweet peptide development, the model trains exclusively on dual-taste peptide data), enabling high-precision prediction of target features. In contrast, 'Multiple Pattern Mode' adopts a more inclusive training strategy (e.g., incorporating sour peptides, sweet peptides, and their combinations for sour-sweet peptide development), providing richer training resources. This strategic design expands the model's exploration of sequence space.

Finally, recognizing that some users may require only peptide toxicity assessment, we separated SpepToxPred as a standalone tool from the TastePepAI platform, providing a dedicated service interface for short peptide toxicity prediction (Fig 6E and 6F). The coordinated deployment of these three platforms not only provides comprehensive technical support for taste peptide research and development but also establishes an open-sharing information platform to advance the field.

### 3. Discussion

In this study, we developed TastePepAI, an automated computational platform that achieves, for the first time, end-to-end automation in taste peptide design. The platform innovatively integrates the core components - the sequence generator LA-VAE and toxicity predictor SpepToxPred – to establish a comprehensive technical framework encompassing molecular design, cluster analysis, safety assessment, and physicochemical property analysis. During development, our systematic analysis of existing taste peptide databases revealed that approximately one-third of peptides possess multiple taste characteristics, with sequence similarities highly overlapping across different taste categories. This complex molecular feature

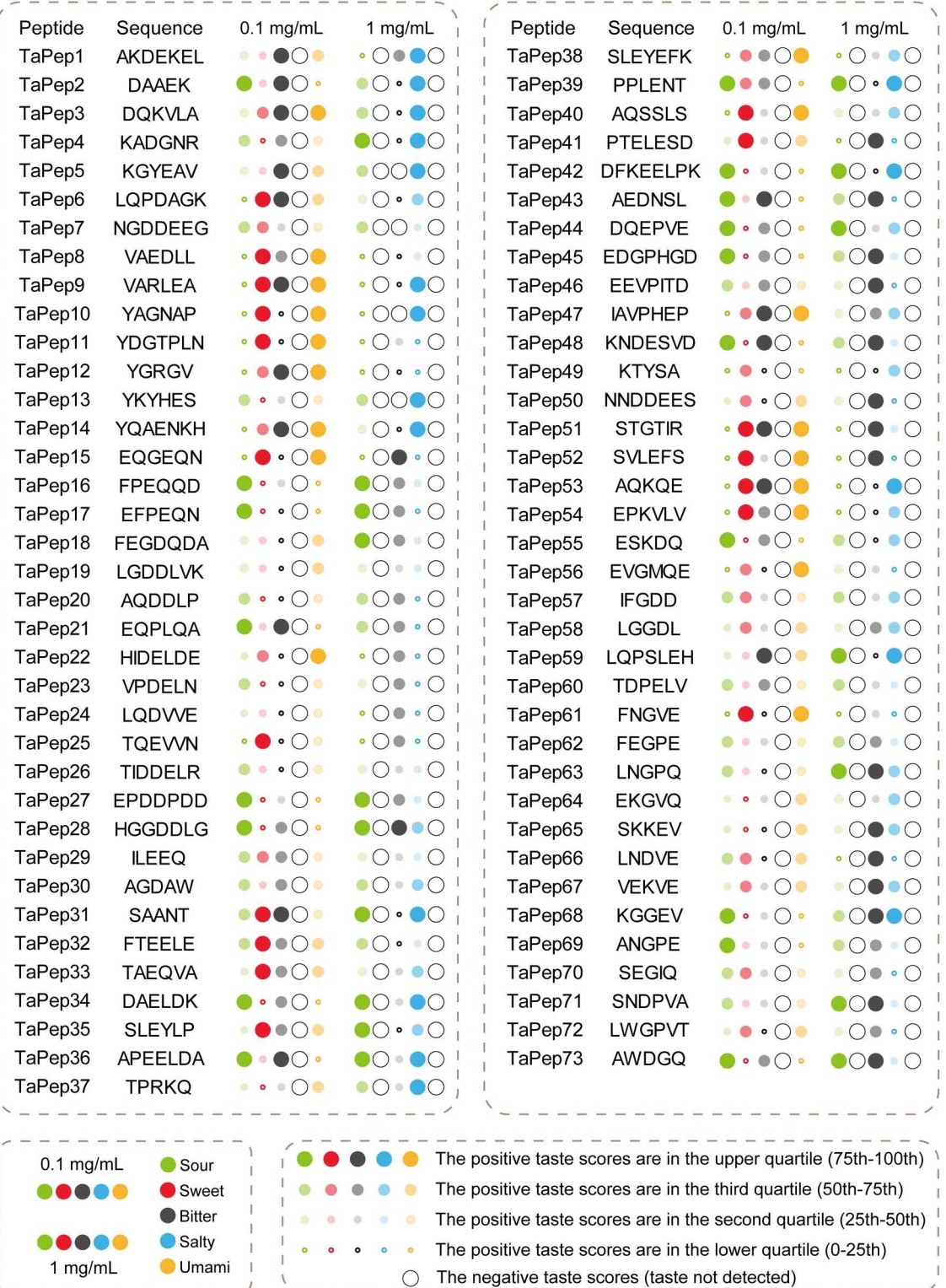

**Fig 5. Electronic tongue analysis reveals concentration-dependent taste profiles of TastePepAI-generated peptides.** Taste characteristics of 73 peptides at two concentrations (0.1 mg/mL and 1 mg/mL). The intensity of each taste modality (sour, sweet, bitter, salty, and umami) is represented by colored dots, where the size reflects the quartile distribution of positive taste scores: large filled dots (75th-100th percentile), medium filled dots

(50th-75th percentile), small filled dots (25th-50th percentile), and tiny dots (0-25th percentile). Large empty circles indicate undetected taste responses. At 0.1 mg/mL concentration, all peptides exhibited sweet and umami characteristics, whereas at 1 mg/mL concentration, a universal salty response was observed across all samples.

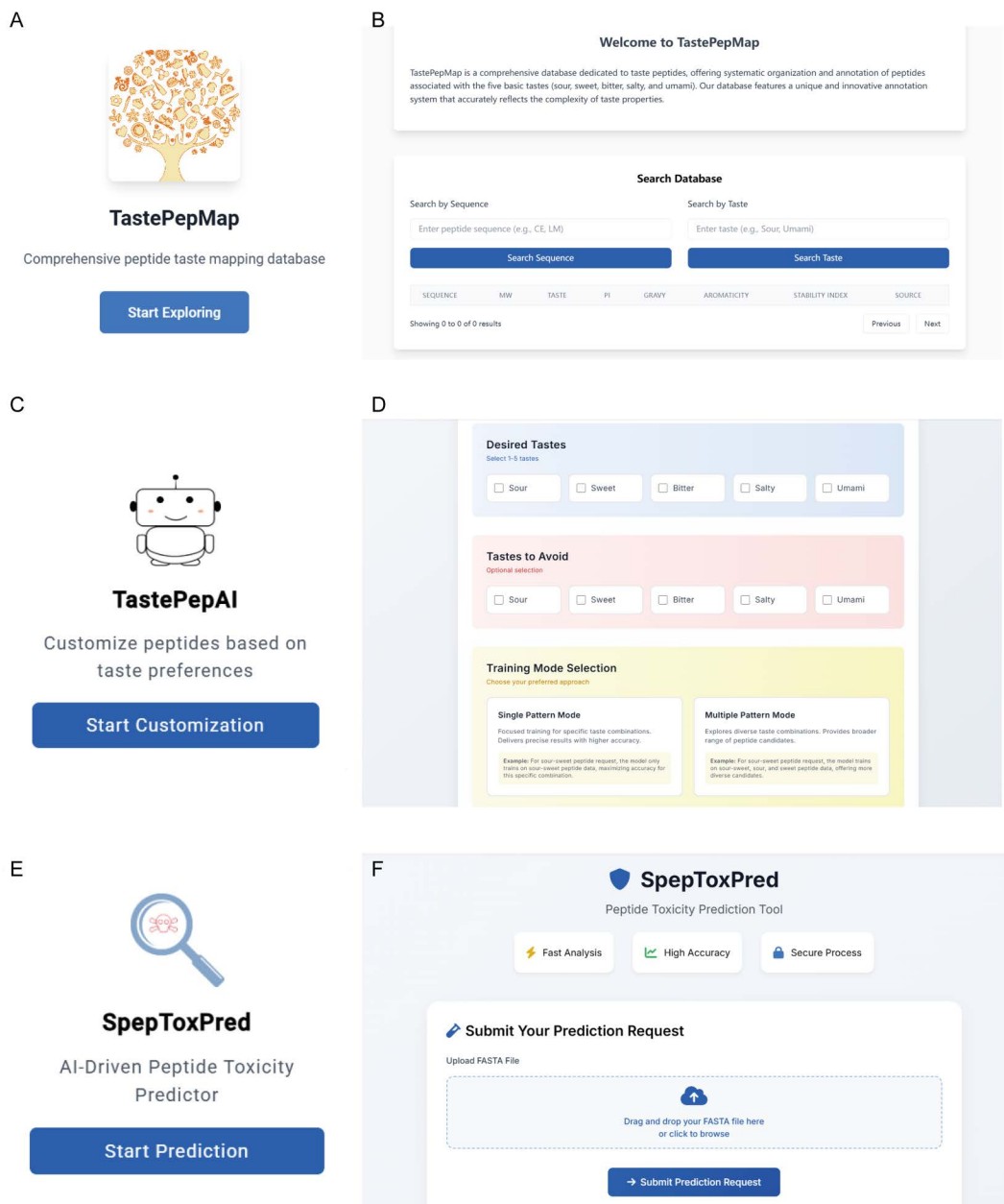

**Fig 6. Development and deployment of integrated open-access platforms for taste peptide research. (A)** Logo and landing page of TastePepMap, a comprehensive database for taste peptides. **(B)** User interface of TastePepMap. **(C)** Logo and entry page of TastePepAI. **(D)** User interface of Taste-PepAI. **(E)** Logo and entry page of SpepToxPred, a tool for AI-driven peptide toxicity prediction. **(F)** User interface of SpepToxPred.

distribution underscores the inherent challenge in taste peptide design: neither traditional experimental screening methods nor existing binary classification approaches can effectively explore such vast and intertwined sequence spaces.

For the sequence generation module, we designed LA-VAE with a loss supervision mechanism through VAE architecture enhancement. To achieve optimal model performance, we introduced a dynamic loss monitoring strategy that precisely tracks and preserves optimal solutions during training, significantly improving sequence generation stability and controllability. This strategy demonstrates excellent data adaptability, automatically adjusting to accommodate dynamic changes in training data, thus providing technical assurance for rapid model deployment in complex data scenarios. Additionally, we constructed a contrastive learning framework for taste characteristics in latent space, enabling precise control of target taste features. This framework not only enhances the expression intensity of desired taste but also effectively suppresses undesired taste characteristics, reducing cross-interference effects in multi-taste peptide design. Furthermore, our intelligent positive/negative sample assignment strategy, combined with the loss supervision mechanism and adaptive extension approach, provides an effective technical approach to mitigate the challenge of limited samples for certain taste combinations by strategically leveraging multi-taste peptides as training resources across different taste patterns and ensuring optimal learning convergence under small dataset conditions. Notably, the design of LA-VAE demonstrates universality, making it applicable not only to taste peptide design but also to other directed molecular sequence optimization tasks.

Regarding safety assessment, the SpepToxPred module significantly improved short peptide toxicity prediction accuracy through systematic feature engineering and model optimization. The module achieved an MCC value exceeding 0.70 on independent test set, representing a 12% improvement over existing best models. More importantly, SpepToxPred's predictions provided reliable safety guidance for subsequent experimental validation, effectively reducing the blindness and resource consumption in experimental screening.

Experimental validation results comprehensively demonstrated TastePepAI's superiority in multi-functional taste peptide design. The platform successfully designed 73 novel multi-taste peptides in a single attempt, with electronic tongue testing confirming their expected sweet, salty, and umami characteristics, while all samples showed no significant toxicity. This breakthrough not only surpasses the total number of similar taste peptides reported in existing literature but also demonstrates the significant advantages of automated computational platforms in complex functional peptide development.

Several directions warrant further exploration. First, taste peptide characterization exhibits strong environmental dependence and subjectivity, where minor changes in testing conditions may lead to perceptual differences [60–62]. This inherent uncertainty could affect training data annotation quality and consequently model learning outcomes, particularly in comprehensive assessment of multiple taste features. Future research will benefit from establishing more standardized taste evaluation systems and refined data annotation mechanisms to enhance model reliability. Second, while current models primarily learn from sequence information, taste peptide functionality may be influenced by multiple factors including conformation and physicochemical properties [63,64]. Integrating this multidimensional feature knowledge holds promise for further improving model prediction accuracy and application value.

## 4. Methods

### 4.1. Ethics approval

All animal experiments were approved by the Institutional Animal Care and Use Committee (IACUC) of Hunan Normal University (approval number: HUNNU2023–435), and the National Institutes of Health guidelines for the performance of animal experiments were followed.

### 4.2. Data collection and preprocessing

**4.2.1. Taste peptide data.** Taste peptide data were comprehensively collected through multiple channels. Initially, data were extracted from specialized taste peptide databases, including BIOPEP-UWM [65] and TastePeptidesDB [35]. Subsequently, we integrated datasets from established taste peptide prediction models (such as Umami-MRNN

[44], VirtuousUmami [66], and IUP-BERT [67]). Additionally, extensive literature searches were conducted on PubMed and Google Scholar using combinations of 'Tastes', 'Sour', 'Sweet', 'Bitter', 'Salty', 'Umami' with 'Peptides' as search keywords. During data preprocessing, sequences containing non-standard amino acid residues were removed, and cases where the same sequence was reported with different taste properties across multiple studies were handled uniformly by annotating them as possessing all reported taste properties. Following initial processing, 1161 peptide sequences were obtained. To reduce data noise and enhance model specificity, sequences of 15 amino acids or longer were further excluded, ultimately resulting in a training dataset comprising 1131 taste peptides.

Taste property annotation presented unique challenges due to the complexity of taste perception. Initially, we designed a five-digit binary annotation system (>abcde) to represent the presence (1) or absence (0) of five basic tastes (sour, sweet, bitter, salty, umami). For instance, peptides with salty and umami tastes were annotated as '>00011', while sweet peptides were labeled as '>01000'. However, this binary approach showed limitations due to variations in sample acquisition methods, taste determination procedures, and experimental conditions including peptide purity and concentration [19,61]. We recognized that the absence of a reported taste in existing studies or its non-detection through specific experimental procedures should not definitively indicate its non-existence. Consequently, we developed a more precise annotation system, marking unconfirmed taste properties with 'x' to indicate 'uncertainty'. Under this improved system, peptides reported to have salty and umami tastes were annotated as '>xxx11', sweet peptides as '>x1xxx', and other taste peptides accordingly. This annotation strategy not only more accurately reflects current knowledge levels but also maintains flexibility for future taste discoveries.

**4.2.2 Toxic peptide data.** To construct a reliable toxicity prediction model, toxic and non-toxic peptide sequences were systematically collected from multiple published model datasets (including ToxGIN [68], ToxinPred 3.0 [55], ToxTeller [69], and ToxIBTL [70]) and professional databases (including Conoserver [71], DRAMP 3.0 [72], CAMPR3 [73], DBAASP v3 [74], and Hemolytik [54]). After removing sequences containing non-standard amino acid residues, the initial dataset comprised 6861 positive samples and 9183 negative samples.

To enhance data quality and ensure model practicality, systematic data preprocessing strategies were implemented: Firstly, considering practical application scenarios (predicting taste peptide toxicity), only sequences not exceeding 25 AA in length were retained. Secondly, redundancy was eliminated using a 90% sequence similarity threshold, maximizing dataset representativeness while minimizing sample redundancy, ultimately yielding 2821 toxic and 4880 non-toxic peptides. To mitigate potential model bias from data imbalance, an equal number of sequences from the negative dataset were randomly sampled to construct a balanced dataset. Finally, the data were split in a 9:1 ratio, resulting in a training set of 2538 toxic and 2538 non-toxic peptides, and a test set of 283 toxic and 283 non-toxic peptides.

## 4.3. Automated workflow of TastePepAI

**4.3.1 Interactive taste feature definition system.** The workflow of TastePepAI initiates with an interactive taste feature definition system. Users define target peptide taste characteristics through a five-digit code comprising '1', '0', and 'x', where '1' indicates desired taste features, '0' represents features to avoid, and 'x' denotes no specific requirement for that taste position. During training data selection, the system processes uncertain labels ('x') through flexible pattern matching: for positions marked as 'x', the model collects peptide sequences containing any taste state ('1', '0', or 'x') at those positions, as 'x' indicates the user has no specific preference or requirement for taste features at that position. For example, users can input '>x1x00' to design a peptide with sweet taste while avoiding salty and umami characteristics. The system offers two operational modes: 'Single Pattern Mode' and 'Multiple Pattern Mode'. In 'Single Pattern Mode', users input a single taste feature code (e.g., '>x1x0x'), and the system filters sequences from the database where the sweet taste position is marked as '1' to form the positive sample set, while simultaneously collecting sequences where the salty taste position is marked as '1' as the negative sample set, thereby achieving precise contrastive learning. In 'Multiple Pattern Mode', users can simultaneously input multiple patterns (e.g., '>x10xx,xx0x1'), and the system constructs positive training sets from sequences where corresponding

positions are marked as '1' and negative training sets from sequences where corresponding positions are marked as '0', then separately merges all matched positive and negative sample sequences to form comprehensive training data. This mode is primarily designed to explore broader sequence-taste relationships, facilitating the model's learning of latent space representations for different taste combinations. Subsequently, the LA-VAE undergoes end-to-end training using the filtered training data to learn latent space representations of target taste features.

**4.3.2. Technical architecture of LA-VAE.** LA-VAE implements a deep neural network architecture where the encoder comprises one-dimensional convolutional layers (Conv1D, filters = 32, kernel_size = 3) and fully connected layers, transforming amino acid sequences into a 2000-dimensional Gaussian latent space through nonlinear mappings. The encoder outputs include mean vectors (z_mean) and log variance vectors (z_log_var) for constructing the posterior distribution of latent variables. The decoder employs a mirror structure, reconstructing sequence probability distributions through inverse mapping. To enhance model generalization, dropout mechanisms are applied post-convolutional layers for regularization, and L1 norm constraints ($\lambda = 0.01$) are imposed on Dense layers.

Model optimization utilizes the Adam algorithm ($\eta = 0.001$), with $Loss_{tol}$ comprising $Loss_{rec}$ and $Loss_{KL}$, where $Loss_{rec}$ undergoes dimensional normalization to balance contributions from sequences of varying lengths. Specifically, in the LA-VAE architecture, the $Loss_{KL}$ compares the encoder-learned posterior distribution q(z|x) with the prior distribution p(z), where the posterior distribution q(z|x) is a parameterized Gaussian distribution defined by the mean vector z_mean and log-variance vector z_log_var output by the encoder, while the prior distribution p(z) is a standard multivariate Gaussian distribution N(0, I). The corresponding loss functions are mathematically formulated as follows. The total loss function of LA-VAE comprises two main components with equal weighting:

$$Loss_{tol} = Loss_{rec} + Loss_{KL}$$

The reconstruction loss employs binary cross-entropy with dimensional scaling to balance the contribution relative to the KL divergence term:

$$Loss_{rec} = \frac{1}{N} \sum_{i=1}^{N} \left[ -\sum_{j=1}^{14} \sum_{k=1}^{21} \left[ x_{i,j,k} \log(\hat{x}_{i,j,k}) + (1 - x_{i,j,k}) \log(1 - \hat{x}_{i,j,k}) \right] \right] \times 294$$

where $N$ is the batch size, $x_{i,j,k}$ represents the one-hot encoded input for the $i$-th sample at position $j$ and amino acid $k$, $\hat{x}_{i,j,k}$ is the corresponding reconstructed output, and the scaling factor $294 = 14 \times 21$ ensures balanced optimization between reconstruction and regularization terms.

The KL divergence loss follows the standard VAE formulation:

$$Loss_{KL} = -\frac{0.5}{N} \sum_{i=1}^{N} \sum_{d=1}^{2000} \left[ 1 + \log\sigma_{i,d}^2 - (\mu_{i,d})^2 - \exp\left(\log\sigma_{i,d}^2\right) \right]$$

where $\mu_{i,d}$ and $\log\sigma_{i,d}^2$ are the mean and log-variance outputs from the encoder for the $d$-th latent dimension of the $i$-th sample, respectively. Both loss components utilize equal weighting (1:1 ratio) to ensure balanced optimization between reconstruction fidelity and latent space regularization.

Based on these loss formulations, the training process of LA-VAE implements dynamic monitoring through customized callback mechanisms, maintaining training states and executing model weight preservation, sequence generation, and distribution visualization operations.

Following successful training, new latent vectors are sampled from the learned 2000-dimensional Gaussian latent space using standard normal distributions, representing potential peptide encodings with target taste properties. These

sampled latent vectors are subsequently processed through the trained decoder network, which transforms each latent representation into probability distributions over the 20 standard amino acids at each of the 14 sequence positions. Final peptide sequences are constructed by selecting the amino acid with highest probability at each position, effectively converting continuous latent representations into discrete peptide sequences suitable for further evaluation and experimental validation.

In taste avoidance mode, the system projects high-dimensional latent space onto two-dimensional manifolds through principal component analysis to characterize positive and negative sample distributions, constructing distance matrices based on Euclidean metrics to provide quantitative criteria for sequence screening.

Specifically, the bilateral distance evaluation strategy computes k-nearest neighbor distances (k = 5) for each generated sequence candidate in the latent space. For every candidate sequence z, the system calculates the average Euclidean distance to the five nearest positive training samples (avg_dist_to_positive) and the five nearest negative training samples (avg_dist_to_negative) when negative samples are available. The 'positive sample affinity' criterion is quantified by minimizing avg_dist_to_positive, ensuring generated sequences maintain close proximity to desired taste characteristics in the latent representation. Conversely, the 'negative sample repulsion' criterion is quantified by maximizing avg_dist_to_negative, ensuring generated sequences maintain sufficient distance from undesired taste properties.

The final sequence selection employs a dual-ranking strategy: sequences are sorted first by avg_dist_to_positive in ascending order (prioritizing positive affinity), then by avg_dist_to_negative in descending order (prioritizing negative repulsion). When both positive and negative samples are used, the system calculates a differential metric (diff_avg_dist_pos_neg = avg_dist_to_positive - avg_dist_to_negative) and retains the top 25% of candidates with the smallest differential values, representing sequences that simultaneously demonstrate strong positive sample affinity and effective negative sample repulsion. This quantitative framework ensures precise control over taste feature generation while maintaining computational efficiency through k-nearest neighbor approximation. Additionally, for LA-VAE hyperparameter tuning, each hyperparameter configuration was run three consecutive times, with the minimum average $Loss_{tol}$ of generated sequences used as the evaluation criterion. Details can be found in S2 Data.

**4.3.3  Hierarchical clustering based on sequence similarity.**  Candidate sequences generated by LA-VAE undergo optimization through an automated hierarchical clustering system. The system initially performs geometry-based screening in latent space: (1) In standard mode, the system calculates Euclidean metrics of sequences in latent space, retaining the 25% of sequences closest to the training manifold; (2) In taste avoidance mode, the system constructs a dual distance metric framework, computing average Euclidean distances ($d_+^-$ and $d_-$) between each generated sequence and its k-nearest neighbors (k = 5) in positive and negative samples, establishing ranking criteria based on distance differential metrics ($\Delta d = d_+^- - d_-$), prioritizing sequences that are simultaneously proximate to target manifolds while distant from avoidance manifolds in latent space.

Filtered sequences are mapped to an undirected weighted network based on sequence homology. Network construction employs an enhanced Needleman-Wunsch global alignment algorithm, quantifying evolutionary distances through scoring matrices (substitution matrix: match = 2.0, mismatch = −1.0) and affine gap penalties (opening = −0.5, extension = −0.1). Normalized alignment scores serve as edge weights, with connectivity subgraphs defined by a similarity threshold (≥70%). For each subgraph, the system selects sequences with highest average similarity as cluster representatives based on node centrality metrics. This graph theory-based clustering strategy achieves automated sequence redundancy elimination while preserving sequence space topology.

**4.3.4.  SpepToxPred toxicity prediction system.**  SpepToxPred, a specialized toxicity prediction system for short peptides (≤25 AA), employs multi-feature fusion and ensemble learning frameworks. At the feature engineering level, the system integrates 20 sequence descriptors: Amino Acid Composition (AAC), Dipeptide Composition (DPC), Tripeptide Composition (TPC), Grouped Amino Acid Composition (GAAC), Grouped Dipeptide Composition (GDPC), Grouped Tripeptide Composition (GTPC), Composition-Transition-Distribution descriptors (CTDC, CTDT, CTDD), Conjoint Triad

descriptors (Ctriad), Enhanced Amino Acid Composition (EAAC), Enhanced Grouped Amino Acid Composition (EGAAC), Composition of k-Spaced Amino Acid Pairs (CKSAAP), Composition of k-Spaced Amino Acid Group Pairs (CKSAAGP), Binary encoding (Binary), BLOSUM62 matrix encoding (BLOSUM62), Dipeptide Deviation Encoding (DDE), Pseudo Amino Acid Composition (PAAC), Amphiphilic Pseudo Amino Acid Composition (APAAC), and Z-scale descriptors (Z-scale). These features undergo StandardScaler normalization and dimensionality reduction optimization through random forest feature selectors.

In model construction, the system integrates nine machine learning algorithms: Random Forest (RF), Extremely Randomized Trees (ERT), Support Vector Machine (SVM), Light Gradient Boosting Machine (LightGBM, LGBM), eXtreme Gradient Boosting (XGBoost, XGB), CatBoost (CAB), K-Nearest Neighbors (KNN), Logistic Regression (LR), and Adaptive Boosting (AdaBoost, ADB). Prediction results are integrated through weighted voting strategies, leveraging complementary advantages of different algorithms in sequence-toxicity pattern recognition.

**4.3.5. Sequence physicochemical property analysis.** Following toxicity prediction, a multidimensional physicochemical property calculation framework based on BioPython [75] and modlamp [76] evaluates sequence characteristics. This framework integrates two sequence analysis tools: (1) BioPython's ProteinAnalysis module calculates Grand Average of Hydropathicity (GRAVY), Isoelectric Point, Net Charge at pH 7.0, Molecular Weight, Aromaticity, Instability Index, secondary structure proportions (α-helix, β-sheet, and turns), and Molar Extinction Coefficients under reduced and oxidized conditions; (2) modlamp's GlobalDescriptor and PeptideDescriptor modules compute Aliphatic Index, Charge Density, Hydrophobic Ratio, Hydrophobic Moment (based on Eisenberg hydrophobicity scale), and solubility-related parameters.

## 4.4. Wet-lab evaluation

**4.4.1. Peptide synthesis.** All peptides used in this study were synthesized by Nanjing Peptide Biotech Co., Ltd. (Nanjing, China) using solid-phase peptide synthesis (SPPS) methodology, with all synthesized peptides achieving a purity exceeding 95%. The high-performance liquid chromatography (HPLC) and mass spectrometry (MS) analytical reports for all 73 samples can be found in S1 File.

**4.4.2. Cell viability assessment.** The cytotoxicity of 73 synthetic peptides was evaluated using four human cell lines: human pancreatic ductal epithelial cells (HPNE), human embryonic kidney cells (HEK293T), human umbilical vein endothelial cells (HUVEC), and human bronchial epithelial cells (BEAS-2B). When cell density reached 70% confluence, the complete culture medium was replaced with serum-free maintenance medium, and peptides were added at a final concentration of 100 μM for 36 h. Control groups received only serum-free maintenance medium without peptides. Cell proliferation was assessed using the CCK-8 assay kit (Solarbio, CA1210) according to the manufacturer's instructions. Detailed cell viability assay results are provided in S3 Data.

**4.3.3. Hemolysis assay.** The hemolytic activity of synthetic peptides was evaluated using red blood cells from 6-week-old BALB/c mice (mRBCs). Freshly isolated mRBCs were washed three times with PBS buffer (8000 rpm, 2 min per wash) to prepare the mRBC suspension. Equal volumes (70 μL) of 200 μM peptide solutions and mRBC suspension were mixed to achieve final concentrations of 100 μM peptide and $1.5 \times 10^8$ cells/mL mRBCs in the reaction system. After incubation with shaking at 37°C for 60 min, samples were centrifuged at 10000 rpm for 5 min, and hemoglobin release was assessed by measuring the absorbance of the supernatant at 490 nm. PBS and 1% Triton X-100 treatments served as 0% and 100% hemolysis controls, respectively. A complete dataset of hemolytic activity measurements is documented in S4 Data.

**4.3.4. Electronic tongue analysis.** Taste characteristics were analyzed using the electronic tongue system E-tongue (Taste-Sensing System SA 402B, Intelligent Sensor Technology Co. Ltd., Japan). The system was equipped with five specific sensor probes for detecting sourness (CA0), saltiness (CT0), bitterness (C00), sweetness (GL1), and umami (AAE) [57,77,78]. Samples were tested at concentrations of 0.1 mg/mL and 1.0 mg/mL. Prior to experiments, sensors

were activated by immersion in a reference solution (30 mM KCl and 0.3 mM tartaric acid) for 24 h. A 30 min self-diagnostic procedure was performed before each measurement to ensure data accuracy. The reference solution served both as a cleaning solution and standard solution for taste signal calibration. According to the manufacturer's instructions, using the reference solution as baseline, the detection thresholds were set at −13 for sourness, −6 for saltiness, and 0 for other taste modalities. Two cleaning solutions were employed for electrode maintenance: (1) 30% ethanol solution containing 100 mM HCl for negative charge reference electrodes, and (2) 30% ethanol solution containing 100 mM KCl and 10 mM KOH for positive charge reference electrodes. All taste measurements were performed at room temperature with four replicates, except for sweetness which was measured five times. The first measurement data were excluded from analysis.

## Supporting information

**S1 Text.** **Fig A. Sequence similarity networks of taste peptides based on global and local alignment algorithms.** (A) Global sequence similarity network constructed using the Needleman-Wunsch algorithm with spring layout algorithm for network visualization optimization. (B) Local sequence similarity network constructed using the Smith-Waterman algorithm with Kamada-Kawai layout algorithm to emphasize local sequence similarities. In both networks, nodes represent individual peptides with size proportional to sequence length, and node colors indicate different taste properties. Edge thickness corresponds to the degree of sequence similarity between peptides. **Fig B. Length distribution and amino acid composition analysis of toxic and non-toxic peptides.** (A) Length distribution of the complete dataset comprising 6861 toxic peptides (red) and 9183 non-toxic peptides (blue). (B) Amino acid frequency distribution in the complete dataset. (C) Length distribution of the filtered dataset (≤25 AA) containing 2821 toxic peptides and 2821 length-matched non-toxic peptides. (D) Amino acid frequency distribution in the filtered dataset (≤25 AA). **Fig C. Length-specific amino acid frequency analysis of toxic peptides.** Frequency distribution analysis of amino acid residues across different sequence length groups in toxic peptides. Each subplot shows how a specific amino acid's frequency (%) varies among peptide sequence sets of different lengths (4–50 AA). The vertical dashed line at 25 AA indicates our sequence length threshold for model development. The mean frequency value is indicated for each residue. **Fig D. Comparative analysis of amino acid frequencies between shorter (≤25 AA) and longer (26–50 AA) toxic peptides.** Frequency distribution (%) of 20 amino acid residues in shorter (≤25 AA, pink) and longer (26–50 AA, blue) toxic peptide sequences. Error bars represent standard errors. Statistical significance levels are indicated ($*p < 0.05$, $**p < 0.01$, $***p < 0.001$). Eighteen amino acids showed significant differences in their frequencies between the two length groups ($p < 0.001$), except for alanine (A, $p = 0.152$) and valine (V, $p = 0.226$), supporting the rationale for length-specific toxicity modeling. **Fig E. Training dynamics and latent space evolution of LA-VAE.** (A, B) Loss curves during model training for (A) positive and (B) negative data across 500 epochs. (C-E) Two-dimensional PCA visualization of the latent space distribution at different training stages. (C) Separation of positive (purple), negative (green), and generated positive (yellow) samples at Step 1 (epoch 10). (D) Intermediate stage showing increased overlap between generated and positive samples at Step 2. (E) Final convergence state demonstrating optimal distribution alignment at Step 3. **Fig F. Sequence and physicochemical property analysis of 10000 TastePepAI-generated peptides.** (A) Amino acid residue frequency distribution (%) comparing positive training samples (red), negative training samples (blue), and generated sequences (orange). (B) Sequence similarity density distribution between generated peptides and positive training samples, calculated using the Needleman-Wunsch algorithm. The majority of generated sequences show less than 50% identity to training samples. (C-L) Box plots comparing physicochemical properties among positive training set (light red), negative set (light blue), and generated sequences (light orange). Box plots show median, quartiles, and whiskers (minimum to maximum). **Fig G. Safety evaluation of 73 peptides.** (A) Cell viability assessment of 73 peptides (100 μM) across four different cell lines: BEAS-2B, HEK293T, HPNE, and HUVEC. (B) Hemolysis rates of 73 peptides (100 μM) using mouse red blood

cells. **Table A. Electronic tongue analysis results of synthetic peptides at 1 mg/mL. Table B. Electronic tongue analysis results of synthetic peptides at 0.1 mg/mL.**
(DOCX)

**S1 Data.** Ensemble model weight optimization results for SpepToxPred, including all tested combinations and corresponding performance metrics.
(XLSX)

**S2 Data.** LA-VAE related hyperparameter settings and testing.
(XLSX)

**S3 Data.** Cell viability assay results for all 73 peptides tested at 100 µM across four human cell lines.
(XLSX)

**S4 Data.** Hemolytic activity measurements for all 73 peptides at 100 µM using mouse red blood cells.
(XLSX)

**S1 File.** HPLC and mass spectrometry analytical reports confirming purity and molecular identity of all 73 synthesized peptides.
(ZIP)

## Acknowlegements

We thank the Bioinformatics Center of Hunan Normal University for providing computer resources, and acknowledge the Key Laboratory of Tea Science of Ministry of Education of Hunan Agricultural University for access to the electronic tongue system E-tongue (Taste-Sensing System SA 402B, Intelligent Sensor Technology Co. Ltd., Japan).

## Author contributions

**Conceptualization:** Jianda Yue, Zhonghua Liu, Zhonghua Liu, Ying Wang.

**Data curation:** Jianda Yue, Tingting Li, Jiawei Xu.

**Formal analysis:** Jianda Yue, Tingting Li.

**Funding acquisition:** Zhonghua Liu, Zhonghua Liu.

**Investigation:** Jianda Yue, Tingting Li, Jian Ouyang, Jiawei Xu, Hua Tan, Zihui Chen.

**Methodology:** Jianda Yue, Tingting Li.

**Project administration:** Ying Wang.

**Resources:** Songping Liang, Zhonghua Liu, Zhonghua Liu, Ying Wang.

**Software:** Jianda Yue, Tingting Li.

**Supervision:** Songping Liang, Zhonghua Liu, Zhonghua Liu, Ying Wang.

**Validation:** Tingting Li, Jian Ouyang, Hua Tan, Changsheng Han, Huanyu Li.

**Visualization:** Jianda Yue.

**Writing – review & editing:** Jianda Yue, Ying Wang.

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
