## [Decision Letter · Decision Letter 0]

11 Sep 2025

TastepepAI:An artificial intelligence platform for taste peptide de novo design

PLOS Computational Biology

Dear Dr. Wang,

Thank you for submitting your manuscript to PLOS Computational Biology. After careful consideration, we feel that it has merit but does not fully meet PLOS Computational Biology's publication criteria as it currently stands. Therefore, we invite you to submit a revised version of the manuscript that addresses the points raised during the review process.

Please submit your revised manuscript within 60 days (November 09, 2025 at 23:59 PM). If you will need more time than this to complete your revisions, please reply to this message or contact the journal office at ploscompbiol@plos.org. Please include the following items when submitting your revised manuscript:

We look forward to receiving your revised manuscript.

Kind regards,

Mohammad Sadegh Taghizadeh, Ph.D.

Academic Editor

PLOS Computational Biology

Arne Elofsson

Section Editor

PLOS Computational Biology

**Journal Requirements:**

1) Please provide an Author Summary. This should appear in your manuscript between the Abstract (if applicable) and the Introduction, and should be 150-200 words long. The aim should be to make your findings accessible to a wide audience that includes both scientists and non-scientists. Sample summaries can be found on our website under Submission Guidelines:

Potential Copyright Issues:

- Figures 1 and 6. Please confirm whether you drew the images / clip-art within the figure panels by hand. If you did not draw the images, please provide (a) a link to the source of the images or icons and their license / terms of use; or (b) written permission from the copyright holder to publish the images or icons under our CC BY 4.0 license. Alternatively, you may replace the images with open source alternatives. See these open source resources you may use to replace images / clip-art:

5) Please ensure that the funders and grant numbers match between the Financial Disclosure field and the Funding Information tab in your submission form. Note that the funders must be provided in the same order in both places as well.

**Reviewers' comments:**

Reviewer's Responses to Questions

**Comments to the Authors:**

Reviewer #1: Yue et al. developed an artificial intelligence platform for designing taste peptides. This manuscript addresses a topic that may be of interest to ploscb as a methods paper. This work provides an artificial intelligence tool for generating target peptides with the desired taste profiles. This approach allows for a reduction in time and resources, thus enabling broader application in the food industry. Although generally well-structured and presented, there are two points that could be further clarified to support this work.

In the results, section 2.2 LA-VAE: a loss-supervised adaptive variational autoencoder with contrastive learning for controlled taste peptide generation paragraph, it's not clear what has been done:

‘To meet the demands for precise control of specific taste properties (e.g., bitter taste elimination40,53) in practical applications, we incorporated a contrastive learning-based properties, enabling bilateral partitioning of training data: sequences with target taste avoidance mechanism into the LA-VAE architecture (Figure 3B) This mechanism allows users to explicitly specify desired (preferred) and avoided (aversive) taste features constitute the positive set, while those with avoided features form the negative set. Through parallel training on these complementary datasets, LA-VAE establishes a structured contrastive representation framework in the latent space.‘

More specifically, what is meant by "bilateral partitioning of the training data"? Furthermore, does this "partitioning" reduce the initial dataset and thus the model training? How does this partitioning ultimately impact decision-making or predictions?

In the results section, 2.4. Design and validation of safe taste peptides via TastePepAI paragraph:

If the training data with "currently no bitterness" were removed, would this have a negative or positive impact on the model's learning?

‘This limited bitterness suppression might be attributed to residual impurities from peptide synthesis (e.g., trifluoroacetic acid, sodium acetate) potentially interfering with sensory evaluation10,56,57, coupled with possible taste evaluation blind spots in training data labeled as ‘currently without bitterness’58,59. Consequently, accurate assessment of bitterness suppression requires further validation.’

How do the authors intend to proceed in validating the suppression of bitterness?

Reviewer #2: The manuscript introduces TastePepAI, an innovative AI framework for customized taste peptide design and safety assessment. The integration of LA-VAE with a toxicity prediction module represents a promising advance in computational peptide engineering and addresses a timely challenge in food science. However, the work suffers from insufficient methodological detail and limited validation.

1. During LA-VAE training, amino acid sequences are reconstructed via an autoencoder to obtain latent representations. However, it remains unclear which two distributions are compared in the KL divergence term. The paper mentions contrastive learning, but does not specify how the contrastive loss is calculated. Please clearly define all loss functions used, together with their weighting scheme in the total loss.

2. The inference stage is insufficiently explained. What is the input during inference? If it is only the desired taste profile (e.g., a character string of taste attributes), how are the corresponding candidate peptide sequences generated from this input?

3. Some hyperparameters lack justification, including latent space dimensionality (2000), K value in KNN (k = 5), and the model weights in SpepToxPred. Please explain the rationale or tuning procedure for these parameters.

4. The criterion that “sequences that simultaneously satisfy both ‘positive sample affinity’ and ‘negative sample repulsion’ are selected as final outputs” is unclear. Please provide a quantitative definition of this criterion.

5. The training dataset contains only 1,131 sequences but encompasses many taste combinations. For certain combinations, the number of available sequences may be very small. Please discuss how the model ensures sufficient positive and negative samples for effective learning.

6. The current validation is insufficient, as the study only demonstrates the generation of bitter-avoidance peptides. It is recommended to include simpler baseline experiments, such as generating peptides with a single taste or with simple dual-taste combinations, and report the success rate.

7. While the experimental verification is commendable, demonstrating the validity of this computational design method based solely on a single experimental set may be insufficient. Ideally, the method should be systematically evaluated using independent test sets and quantitative metrics (e.g., sequence recovery rates analogous to protein design studies). If this is challenging, at minimum, a confidence or uncertainty measure for the generated sequences should be provided to allow users to assess their reliability.

8. The term “losstol” appears multiple times but is not defined. Please provide a clear explanation.

9. The description of the dataset (sources, size, preprocessing and filtering criteria) should be provided in the Methods section. The Results section can present statistical characteristics or distributional analyses.

10. Regarding uncertain taste labels (“x”), the paper introduces this notation but does not explain how the model processes such labels.

11. Ensure consistent terminology when describing taste categories (e.g., always use the plural form: “sweet, salty, and umami tastes”).

Reviewer #3: The paper proposes a computational pipeline for the generation of taste peptides. The main components of the methodology are loss-supervised adaptive variational autoencoder (LA-VAE) for the generation of peptides, combined with a toxicity prediction model. The approach has been validated in vitro (by testing synthesized peptides on cell cultures) and through an electronic tongue. A collection of generated taste peptides has been made available through the development of a public database.

The paper is interesting, methodologically sound and well written. The approach is original and represents a significant advancement of the state of the art. Consequently, my overall opinion on the paper is positive.

I only have a few minor comments:

- The github repository with the source code is available, but the web pages at the domain http://www.wang-subgroup.com/ (which seem to contain a lot of necessary stuff) are not available. When I try to connect (from Europe) I get an error page with text in (probably) Chinese language. Maybe this is a small technical problem, easy to fix (this is why I suggest "minor" revision), but the availability of these web pages is crucial for the acceptance of this paper

- Why the "all_test.fast" file in the github repository contains only very short sequences?

- In the abstract: "to efficiently optimizes" -> "to efficiently optimize" and also "facilitates" -> "facilitate"

- In the introduction: "Here, this paper" sounds weird

**Have the authors made all data and (if applicable) computational code underlying the findings in their manuscript fully available?**

Reviewer #1: None

Reviewer #2: Yes

Reviewer #3: Yes

PLOS authors have the option to publish the peer review history of their article (what does this mean? ). If published, this will include your full peer review and any attached files.

**Do you want your identity to be public for this peer review?** For information about this choice, including consent withdrawal, please see our Privacy Policy .

Reviewer #1: No

Reviewer #2: No

Reviewer #3: No

**Figure resubmission:**

**Reproducibility:**



---

## [Decision Letter · Decision Letter 1]

5 Oct 2025

PCOMPBIOL-D-25-01523R1

TastepepAI:An artificial intelligence platform for taste peptide de novo design

PLOS Computational Biology

Dear Dr. Wang,

Thank you for submitting your manuscript to PLOS Computational Biology. After careful consideration, we feel that it has merit but does not fully meet PLOS Computational Biology's publication criteria as it currently stands. Therefore, we invite you to submit a revised version of the manuscript that addresses the points raised during the review process.

Please submit your revised manuscript within 30 days (November 01, 2025, at 23:59). If you will need more time than this to complete your revisions, please reply to this message or contact the journal office at ploscompbiol@plos.org. Please include the following items when submitting your revised manuscript:

We look forward to receiving your revised manuscript.

Kind regards,

Mohammad Sadegh Taghizadeh, Ph.D.

Academic Editor

PLOS Computational Biology

Arne Elofsson

Section Editor

PLOS Computational Biology

**Journal Requirements:**

1) Please revise your current Competing Interest statement to the standard "The authors have declared that no competing interests exist."

2) Please ensure that the links provided in the Data Availability statement are working. 

**Reviewers' comments:**

Reviewer's Responses to Questions

Reviewer #1: Yue et al. responded appropriately to the recommendations and revised the manuscript to clarify critical aspects. They also clarified the impact on decision-making or model predictions under other conditions as requested.

Reviewer #2: The authors have addressed my concerns regarding the inference stage by clarifying the input format and providing details on how training data are selected and processed based on user-defined patterns. However, one aspect remains insufficiently clear. In the revised text, the inference stage is described as: “Subsequently, the LA-VAE undergoes end-to-end training using the filtered training data to learn latent space representations of target taste features.” While this explains how the model learns latent representations, it does not specify how latent vectors are subsequently sampled and decoded into candidate peptide sequences. I suggest that the authors provide additional details on the inference workflow, particularly how sampling is performed in the latent space and how these samples are transformed into peptide sequences.

Reviewer #3: The authors addressed my comments in a satisfactory way in the new version of the paper.

**Have the authors made all data and (if applicable) computational code underlying the findings in their manuscript fully available?**

Reviewer #1: Yes

Reviewer #2: Yes

Reviewer #3: Yes

PLOS authors have the option to publish the peer review history of their article (what does this mean? ). If published, this will include your full peer review and any attached files.

**Do you want your identity to be public for this peer review?** For information about this choice, including consent withdrawal, please see our Privacy Policy .

Reviewer #1: No

Reviewer #2: No

Reviewer #3: No

**Figure resubmission:**

**Reproducibility:**



---

## [Decision Letter · Decision Letter 2]

9 Oct 2025

Dear Dr. Wang,

We are pleased to inform you that your manuscript 'TastepepAI:An artificial intelligence platform for taste peptide de novo design' has been provisionally accepted for publication in PLOS Computational Biology.

Best regards,

Mohammad Sadegh Taghizadeh, Ph.D.

Academic Editor

PLOS Computational Biology

Arne Elofsson

Section Editor

PLOS Computational Biology

Reviewer's Responses to Questions

**Comments to the Authors:**

Reviewer #2: The authors have addressed all my concerns. I have no more questions.

**Have the authors made all data and (if applicable) computational code underlying the findings in their manuscript fully available?**

Reviewer #2: Yes

PLOS authors have the option to publish the peer review history of their article (what does this mean? ). If published, this will include your full peer review and any attached files.

**Do you want your identity to be public for this peer review?** For information about this choice, including consent withdrawal, please see our Privacy Policy .

Reviewer #2: No

---

## [Editor Report · Acceptance letter]

PCOMPBIOL-D-25-01523R2

TastepepAI:An artificial intelligence platform for taste peptide de novo design

Dear Dr Wang,

I am pleased to inform you that your manuscript has been formally accepted for publication in PLOS Computational Biology. Your manuscript is now with our production department and you will be notified of the publication date in due course.

With kind regards,

Anita Estes
